# Learning to Self-Train for Semi-Supervised Few-Shot Classification

**Xinzhe Li**[1*]    **Qianru Sun**[2†]    **Yaoyao Liu**[3*]    **Shibao Zheng**[1†]

**Qin Zhou**[4]    **Tat-Seng Chua**[5]    **Bernt Schiele**[6]

[1]Shanghai Jiao Tong University [2]Singapore Management University [3]Tianjin University [4]Alibaba Group
[5]National University of Singapore [6]Max Planck Institute for Informatics, Saarland Informatics Campus

## Abstract

Few-shot classification (FSC) is challenging due to the scarcity of labeled training data (e.g. only one labeled data point per class). Meta-learning has shown to achieve promising results by learning to initialize a classification model for FSC. In this paper we propose a novel semi-supervised meta-learning method called **learning to self-train (LST)** that leverages unlabeled data and specifically meta-learns how to cherry-pick and label such unsupervised data to further improve performance. To this end, we train the LST model through a large number of semi-supervised few-shot tasks. On each task, we train a few-shot model to predict pseudo labels for unlabeled data, and then iterate the self-training steps on labeled and pseudo-labeled data with each step followed by fine-tuning. We additionally learn a soft weighting network (SWN) to optimize the self-training weights of pseudo labels so that better ones can contribute more to gradient descent optimization. We evaluate our LST method on two ImageNet benchmarks for semi-supervised few-shot classification and achieve large improvements over the state-of-the-art method. Code is at github.com/xinzheli1217/learning-to-self-train.

## 1 Introduction

Today's deep neural networks require large amounts of labeled data for supervised training and best performance [39, 8, 30]. Their potential applications to the small-data regimes are thus limited. There has been growing interest in reducing the required amount of data, e.g. to only 1-shot [12]. One of the most powerful methods is meta-learning that transfers the *experience* learned from similar tasks to the target task [3]. Among different meta strategies, gradient descent based methods are particularly promising for today's neural networks [3, 32, 27]. Another intriguing idea is to additionally use unlabeled data. Semi-supervised learning using unlabeled data with a relatively small set of labeled ones has obtained good performance on standard datasets [21, 20]. A classic, intuitive and simple method is e.g. self-training. It first trains a supervised model with labeled data, and then enlarges the labeled set based on the most confident predictions (called pseudo labels) on unlabeled data [40, 35, 20]. It can outperform regularization based methods [17, 6, 9], especially when labeled data is scarce.

The focus of this paper is thus on the semi-supervised few-shot classification (SSFSC) task. Specifically, there are few labeled data and a much larger amount of unlabeled data for training classifiers. To tackle this problem, we propose a new SSFSC method called **learning to self-train (LST)** that successfully embeds a well-performing semi-supervised method, i.e. self-training, into the meta

gradient descent paradigm. However, this is non-trivial, as directly applying self-training recursively may result in gradual drifts and thus adding noisy pseudo-labels [41]. To address this issue, we propose both to meta-learn a soft weighting network (SWN) to automatically reduce the effect of noisy labels, as well as to fine-tune the model with only labeled data after every self-training step.

Specifically, our LST method consists of inner-loop self-training (for one task) and outer-loop meta-learning (over all tasks). LST meta-learns both *to initialize a self-training model* and *how to cherry-pick from noisy labels* for each task. An inner loop starts from the meta-learned initialization by which a task-specific model can be fast adapted with few labeled data. Then, this model is used to predict pseudo labels, and labels are weighted by the meta-learned soft weighting network (SWN). Self-training consists of re-training using weighted pseudo-labeled data and fine-tuning on few labeled data. In the outer loop, the performance of these meta-learners are evaluated via an independent validation set, and parameters are optimized using the corresponding validation loss.

In summary, our LST method learns to accumulate *self-supervising experience* from SSFSC tasks in order to quickly adapt to a new few-shot task. **Our contribution** is three-fold. (i) A novel self-training strategy that prevents the model from drifting due to label noise and enables robust recursive training. (ii) A novel meta-learned cherry-picking method that optimizes the weights of pseudo labels particularly for fast and efficient self-training. (iii) Extensive experiments on two versions of ImageNet benchmarks – miniImageNet [36] and tieredImageNet [24], in which our method achieves top performance.

## 2 Related works

**Few-shot classification (FSC).** Most FSC works are based on supervised learning. They can be roughly divided into four categories: (1) data augmentation based methods [15, 29, 37, 38] generate data or features in a conditional way for few-shot classes; (2) metric learning methods [36, 31, 33] learn a similarity space of image features in which the classification should be efficient with few examples; (3) memory networks [18, 28, 22, 16] design special networks to record training "experience" from seen tasks, aiming to generalize that to the learning of unseen ones; and (4) gradient descent based methods [3, 4, 1, 23, 11, 7, 42, 32, 14] learn a meta-learner in the outer loop to initialize a base-learner for the inner loop that is then trained on a novel few-shot task. In our LST method, the outer-inner loop optimization is based on the gradient descent method. Different to previous works, we propose a novel meta-learner that assigns weights to pseudo-labeled data, particularly for semi-supervised few-shot learning.

**Semi-supervised learning (SSL).** SSL methods aim to leverage unlabeled data to obtain decision boundaries that better fit the underlying data structure [20]. The $\Pi$-Model applies a simple consistency regularization [9], e.g. by using dropout, adding noise and data augmentation, in which data is automatically "labeled". Mean Teacher is more stable version of the $\Pi$-Model by making use of a moving average technique [34]. Visual Adversarial Training (VAT) regularizes the network against the adversarial perturbation, and it has been shown to be an effective regularization [17]. Another popular method is Entropy Minimization that uses a loss term to encourage low-entropy (more confident) predictions for unlabeled data, regardless of their real classes [6]. Pseudo-labeling is a self supervised learning method that relies on the predictions of unlabeled data, i.e. pseudo labels [2]. It can outperform regularization based methods, especially when labeled data is scarce [20] as in our envisioned setting. We thus use this method in our inner loop training.

**Semi-supervised few-shot classification (SSFSC).** Semi-supervised learning on FSC tasks aims to improve the classification accuracy by adding a large number of unlabeled data in training. Ren *et al*. proposed three semi-supervised variants of ProtoNets [31], basically using Soft $k$-Means method to tune clustering centers with unlabeled data. A more recent work used the transductive propagation network (TPN) [13] to propagate labels from labeled data to unlabeled ones, and meta-learned the key hyperparameters of TPN. Differently, we build our method based on the simple and classical self-training [40] and meta gradient descent method [3, 32] without requiring a new design of a semi-supervised network. Rohrbach *et al*. [25] proposed to further leverage external knowledge, such as the semantic attributes of categories, to solve not only few-shot but also zero-shot problems. Similarly, we expect further gains of our approach when using similar external knowledge in our future work.

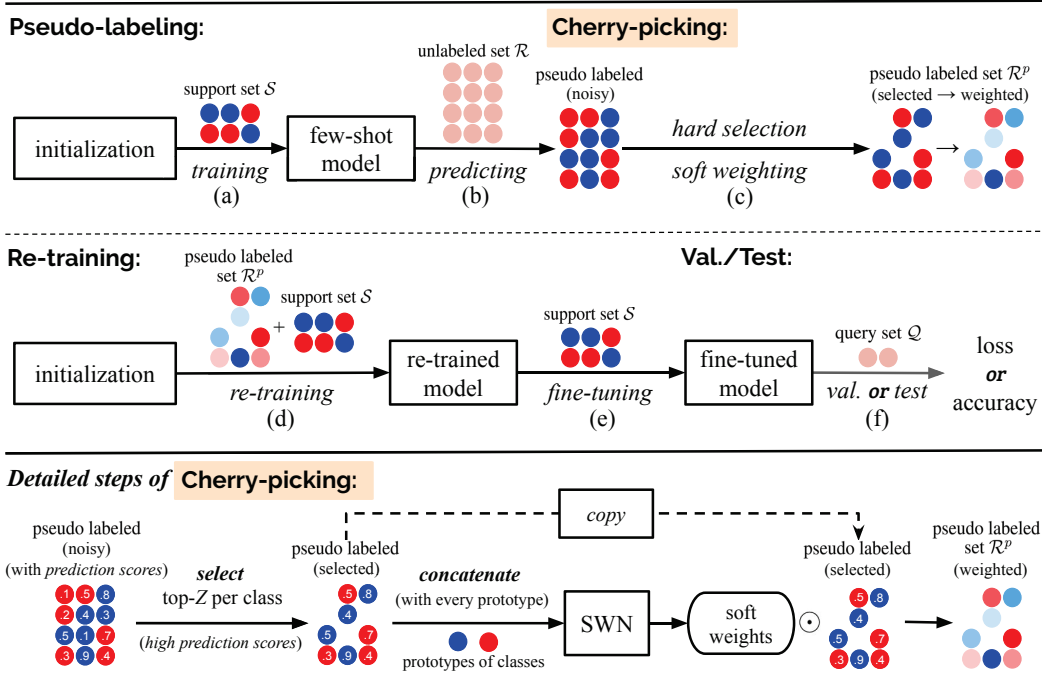

Figure 1: The pipeline of the proposed **LST** method on a single (2-class, 3-shot) task. The prototype of a class is the mean feature in the class, and SWN is the soft weighting network whose optimization procedure is given in Figure 2 and Section 4.2.

## 3  Problem definition and denotation

In conventional few-shot classification (FSC), each task has a small set of labeled training data called support set $\mathcal{S}$, and another set of unseen data for test, called query set $\mathcal{Q}$. Following [24], we denote another set of unlabeled data as $\mathcal{R}$ to be used for semi-supervised learning (SSL). $\mathcal{R}$ may or may not contain data of distracting classes (not included in $\mathcal{S}$).

Our method follows the uniform episodic formulation of meta-learning [36] that is different to traditional classification in three aspects. (1) Main phases are meta-train and meta-test (instead of train and test), each of which includes training (and self-training in our case) and test. (2) Samples in meta-train and meta-test are not datapoints but episodes (SSFSC tasks in our case). (3) Meta objective is not to classify unseen datapoints but to fast adapt the classifier on a new task. Let's detail the denotations. Given a dataset $\mathcal{D}$ for meta-train, we first sample SSFSC tasks $\{\mathcal{T}\}$ from a distribution $p(\mathcal{T})$ such that each $\mathcal{T}$ has few samples from few classes, e.g. 5 classes and 1 sample per class. $\mathcal{T}$ has a support set $\mathcal{S}$ plus an unlabeled set $\mathcal{R}$ (with a larger number of samples) to train a task-specific SSFSC model, and a query set $\mathcal{Q}$ to compute a validation loss used to optimize meta-learners. For meta-test, given an unseen new dataset $\mathcal{D}_{un}$, we sample a new SSFSC task $\mathcal{T}_{un}$. "Unseen" means there is no overlap of image classes (including distracting classes) between meta-test and meta-train tasks . We first initialize a model and weight pseudo labels for this unseen task, then self-train the model on $\mathcal{S}_{un}$ and $\mathcal{R}_{un}$. We evaluate the self-training performance on a query set $\mathcal{Q}_{un}$. If we have multiple unseen tasks, we report average accuracy as the final evaluation.

## 4  Learning to self-train (LST)

The computing flow of applying LST to a single task is given in Figure 1. It contains *pseudo-labeling* unlabeled samples by a few-shot model pre-trained on the support set; *cherry-picking* pseudo-labeled samples by hard selection and soft weighting; *re-training* on picked "cherries", followed by a fine-tuning step; and the *final test* on a query set. On a meta-train task, *final test* acts as a validation to output a loss for optimizing meta-learned parameters of LST, as shown in Figure 2.

## 4.1 Pseudo-labeling & cherry-picking unlabeled data

**Pseudo-labeling.** This step deploys a supervised few-shot method to train a task-specific classifier $\theta$ on the support set $\mathcal{S}$. Pseudo labels of the unlabeled set $\mathcal{R}$ are then predicted by $\theta$. Basically, we can use different methods to learn $\theta$. We choose a top-performing one – meta-transfer learning (MTL) [32] (for fair comparison we also evaluate this method as a component of other semi-supervised methods [24, 13]) that is based on simple and elegant gradient descent optimization [3]. In the outer loop meta-learning, MTL learns scaling and shifting parameters $\Phi_{ss}$ to fast adapt a large-scale pre-trained network $\Theta$ (e.g. for 64 classes and 600 images per class on miniImageNet [36]) to a new learning task. In the inner loop base-learning, MTL takes the last fully-connected layer as classifier $\theta$ and trains it with $\mathcal{S}$.

In the following, we detail the pseudo-labeling process on a task $\mathcal{T}$. Given the support set $\mathcal{S}$, its loss is used to optimize the task-specific base-learner (classifier) $\theta$ by gradient descent:

$$\theta_t \leftarrow \theta_{t-1} - \alpha \nabla_{\theta_{t-1}} L\big(\mathcal{S}; [\Phi_{ss}, \theta_{t-1}]\big), \tag{1}$$

where $t$ is the iteration index and $t \in \{1, ..., T\}$. The initialization $\theta_0$ is given by $\theta'$ which is meta-learned (see Section 4.2). Once trained, we feed $\theta_T$ with unlabeled samples $\mathcal{R}$ to get pseudo labels $Y^{\mathcal{R}}$ as follows,

$$Y^{\mathcal{R}} = f_{[\Phi_{ss}, \theta_T]}(\mathcal{R}), \tag{2}$$

where $f$ indicates the classifier function with parameters $\theta_T$ and feature extractor with parameters $\Phi_{ss}$ (the frozen $\Theta$ is omitted for simplicity).

**Cherry-picking.** As directly applying self-training on pseudo labels $Y^{\mathcal{R}}$ may result in gradual drifts due to label noises, we propose two countermeasures in our LST method. The first is to meta-learn the SWN that automatically reweighs the data points to up-weight the more promising ones and down-weighs the less promising ones, i.e. learns to cherry-pick. Prior to this step we also perform hard selection to only use the most confident predictions [35]. The second countermeasure is to fine-tune the model with only labeled data (in $\mathcal{S}$) after every self-training step (see Section 4.2).

Specifically, we refer to the confident scores of $Y^{\mathcal{R}}$ to pick-up the top $Z$ samples per class. Therefore, we have $ZC$ samples from $C$ classes in this pseudo-labeled dataset, namely $\mathcal{R}^p$. Before feeding $\mathcal{R}^p$ to re-training, we compute their soft weights by a meta-learned soft weighting network (SWN), in order to reduce the effect of noisy labels. These weights should reflect the relations or distances between pseudo-labeled samples and the representations of $C$ classes. We refer to a supervised method called RelationNets [33] which makes use of relations between support and query samples for traditional few-shot classification.

First, we compute the prototype feature of each class by averaging the features of all its samples. In the 1-shot case, we use the unique sample feature as prototype. Then, given a pseudo-labeled sample $(x_i, y_i) \in \mathcal{R}^p$, we concatenate its feature with $C$ prototype features, then feed them to SWN. The weight on the $c$-th class is as follows,

$$w_{i,c} = f_{\Phi_{swn}}\left(\left[f_{\Phi_{ss}}(x_i); \frac{\sum_k f_{\Phi_{ss}}(x_{c,k})}{K}\right]\right), \tag{3}$$

where $c$ is the class index and $c \in [1, ..., C]$, $k$ is the sample index in one class and $k \in [1, ..., K]$, $x_{c,k} \in \mathcal{S}$, and $\Phi_{swn}$ denotes the parameters of SWN whose optimization procedure is given in Section 4.2. Note that $\{w_{i,c}\}$ have been normalized over $C$ classes through a softmax layer in SWN.

## 4.2 Self-training on cherry-picked data

As shown in Figure 2 (inner loop), our self-training contains two main stages. The first stage contains a few steps of re-training on the pseudo-labeled data $\mathcal{R}^p$ in conjunction with support set $\mathcal{S}$, and the second are fine-tuning steps with only $\mathcal{S}$.

We first initialize the classifier parameters as $\theta_0 \leftarrow \theta'$, where $\theta'$ is meta-optimized by previous tasks in the outer loop. We then update $\theta_0$ by gradient descent on $\mathcal{R}^p$ and $\mathcal{S}$. Assuming there are $T$ iterations, re-training takes the first $1 \sim m$ iterations and fine-tuning takes the rest $m + 1 \sim T$. For $t \in \{1, ..., m\}$, we have

$$\theta_t \leftarrow \theta_{t-1} - \alpha \nabla_{\theta_{t-1}} L\big(\mathcal{S} \cup \mathcal{R}^p; [\Phi_{swn}, \Phi_{ss}, \theta_{t-1}]\big), \tag{4}$$

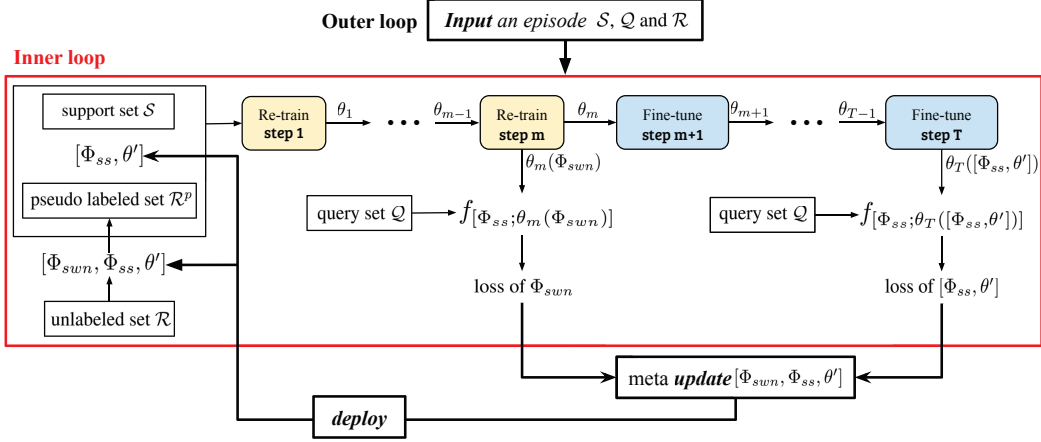

Figure 2: Outer-loop and inner-loop training procedures in our LST method. The inner loop in the red box contains the $m$ steps of re-training (with $\mathcal{S}$ and $\mathcal{R}^p$) and $T - m$ steps of fine-tuning (with only $\mathcal{S}$). In recursive training, the fine-tuned $\theta_T$ replaces the initial MTL learned $\theta_T$ (see Section 4.1) for the pseudo-labeling at the next stage.

where $\alpha$ is the base learning rate. $L$ denotes the classification losses that are different for samples from different sets, as follows,

$$L\big(\mathcal{S} \cup \mathcal{R}^p; [\Phi_{swn}, \Phi_{ss}, \theta_t]\big) = \left\{ \begin{array}{l} L_{ce}\big(f_{[\Phi_{swn}, \Phi_{ss}, \theta_t]}(x_i), y_i\big), \text{ if } (x_i, y_i) \in \mathcal{S}, \\ L_{ce}\big(\mathbf{w}_i \odot f_{[\Phi_{swn}, \Phi_{ss}, \theta_t]}(x_i), y_i\big), \text{ if } (x_i, y_i) \in \mathcal{R}^p, \end{array} \right. \quad (5)$$

where $L_{ce}$ is the cross-entropy loss. It is computed in a standard way on $\mathcal{S}$. For a pseudo-labeled sample in $\mathcal{R}^p$, its predictions are weighted by $\mathbf{w}_i = \{w_{i,c}\}_{c=1}^{C}$ before going into the softmax layer. For $t \in \{m + 1, ..., T\}$, $\theta_t$ is fine-tuned on $\mathcal{S}$ as

$$\theta_t \leftarrow \theta_{t-1} - \alpha \bigtriangledown_{\theta_{t-1}} L(\mathcal{S}; [\Phi_{swn}, \Phi_{ss}, \theta_{t-1}]). \quad (6)$$

**Iterating self-training using fine-tuned model.** Conventional self-training often follows an iterative procedure, aiming to obtain a gradually enlarged labeled set [40, 35]. Similarly, our method can be iterated once a fine-tuned model $\theta_T$ is obtained, i.e. to use $\theta_T$ to predict better pseudo labels on $\mathcal{R}$ and re-train $\theta$ again. There are two scenarios: (1) the size of $\mathcal{R}$ is small, e.g. 10 samples per class, so that self-training can only be repeated on the same data; and (2) that size is infinite (at least big enough, e.g. 100 samples per class), we can split it into multiple subsets (e.g. 10 subsets and each one has 10 samples) and do the recursive learning each time on a new subset. In this paper, we consider the second scenario. We also validate in experiments that first splitting subsets and then recursive training is better than using the whole set for one re-training round.

**Meta-optimizing $\Phi_{swn}$, $\Phi_{ss}$ and $\theta'$.** Gradient descent base methods typically use $\theta_T$ to compute the validation loss on query set $\mathcal{Q}$ used for optimizing meta-learner [32, 3]. In this paper, we have multiple meta-learners with the parameters $\Phi_{swn}$, $\Phi_{ss}$ and $\theta'$. We propose to update them by the validation losses calculated at different self-training stages, aiming to optimize them particularly towards specific purposes. $\Phi_{ss}$ and $\theta'$ work for feature extraction and final classification affecting on the whole self-training. We optimize them by the loss of the final model $\theta_T$. While, $\Phi_{swn}$ produces soft weights to refine the re-training steps, and its quality should be evaluated by re-trained classifier $\theta_m$. We thus use the loss of $\theta_m$ to optimize it. Two optimization functions are as follows,

$$\Phi_{swn} \quad =: \quad \Phi_{swn} - \beta_1 \bigtriangledown_{\Phi_{swn}} L(\mathcal{Q}; [\Phi_{swn}, \Phi_{ss}, \theta_m]), \quad (7)$$

$$[\Phi_{ss}, \theta'] \quad =: \quad [\Phi_{ss}, \theta'] - \beta_2 \bigtriangledown_{[\Phi_{ss}, \theta']} L(\mathcal{Q}; [\Phi_{swn}, \Phi_{ss}, \theta_T]), \quad (8)$$

where $\beta_1$ and $\beta_2$ are meta learning rates that are manually set in experiments.

# 5   Experiments

We evaluate the proposed **LST** method in terms of few-shot image classification accuracy in semi-supervised settings. Below we describe the two benchmarks we evaluate on, details of settings, comparisons to state-of-the-art methods, and an ablation study.

## 5.1 Datasets and implementation details

**Datasets.** We conduct our experiments on two subsets of ImageNet [26]. **miniImageNet** was firstly proposed by Vinyals *et al*. [36] and has been widely used in supervised FSC works [3, 23, 32, 27, 7, 5], as well as semi-supervised works [13, 24]. In total, there are 100 classes with 600 samples of $84 \times 84$ color images per class. In the uniform setting, these classes are divided into 64, 16, and 20 respectively for meta-train, meta-validation, and meta-test. **tieredImageNet** was proposed by Ren *et al*. [24]. It includes a larger number of categories, 608 classes, than miniImageNet. These classes are from 34 super-classes which are divided into 20 for meta-train (351 classes), 6 for meta-validation (97 classes), and 8 for meta-test (160 classes). The average image number per class is 1281, which is much bigger than that on miniImageNet. All images are resized to $84 \times 84$. On both datasets, we follow the semi-supervised task splitting method used in previous works [24, 13]. We consider the 5-way classification, and sample 5-way, 1-shot (5-shot) task to contain 1 (5) samples as the support set $\mathcal{S}$ and 15 samples (a uniform number) samples as the query set $\mathcal{Q}$. Then, on the 1-shot (5-shot) task, we have 30 (50) unlabeled images per class in the unlabeled set $\mathcal{R}$. After hard selection, we filter out 10 (20) samples and only use the rest 20 (30) confident ones to do soft weighting and then re-training. In the recursive training, we use a larger unlabeled data pool containing 100 samples from which each iteration we can sample a number of samples, i.e. 30 (50) samples for 1-shot (5-shot).

**Network architectures** of $\Theta$ and $\Phi_{ss}$ are based on ResNet-12 (see details of MTL [32]) which consist of 4 residual blocks and each block has 3 CONV layers with $3 \times 3$ kernels. At the end of each block, a $2 \times 2$ max-pooling layer is applied. The number of filters starts from 64 and is doubled every next block. Following residual blocks, a mean-pooling layer is applied to compress the feature maps to a 512-dimension embedding. The architecture of SWN consists of 2 CONV layers with $3 \times 3$ kernels in 64 filters, followed by 2 FC layers with the dimensionality of 8 and 1, respectively.

**Hyperparameters**. We follow the settings used in MTL [32]. Base-learning rate $\alpha$ (in Eq. 1, Eq. 4 and Eq. 6) is set to 0.01. Meta-learning rates $\beta_1$ and $\beta_2$ (in Eq. 7 and Eq. 8) are set to 0.001 initially and decay to the half value every $1k$ meta iterations until a minimum value 0.0001 is reached. We use a meta-batch size of 2 and run $15k$ meta iterations. In recursive training, we use 6 (3) recursive stages for 1-shot (5-shot) tasks. Each recursive stage contains 10 re-training and 30 fine-tuning steps.

**Comparing methods.** In terms of SSFSC, we have two methods, namely Soft Masked $k$-Means [24] and TPN [13] to compare with. Their original models used a shallow, i.e. 4CONV [3] trained from scratch, for feature extraction. For fair comparison, we implement the MTL as a component of their models in order to use deeper nets and pre-trained models which have been proved better. In addition, we run these experiments using the maximum budget of unlabeled data, i.e. 100 samples per class. We also compare to the state-of-the-art supervised FSC models which are closely related to ours. They are based on either data augmentation [15, 29] or gradient descent [3, 23, 7, 5, 42, 19, 27, 32, 10].

**Ablative settings.** In order to show the effectiveness of our LST method, we design following settings belonging to two groups: with and without meta-training. Following are the detailed ablative settings. *no selection* denotes the baseline of once self-training without any selection of pseudo labels. *hard* denotes hard selection of pseudo labels. *hard* with meta-training means meta-learning only $[\Phi_{ss}, \theta_T]$. *soft* denotes soft weighting on selected pseudo labels by meta-learned SWN. *recursive* applies multiple iterations of self-training based on fine-tuned models, see Section 4.2. Note that this *recursive* is only for the meta-test task, as the meta-learned SWN can be repeatedly used. We also have a comparable setting to *recursive* called *mixing* in which we mix all unlabeled subsets used in *recursive* and run only one re-training round (see the last second paragraph of Section 4.2).

## 5.2 Results and analyses

We conduct extensive experiments on semi-supervised few-shot classification. In Table 1, we present our results compared to the state-of-the-art FSC methods, respectively on miniImageNet and tieredImageNet. In Table 2, we provide experimental results for ablative settings and comparisons with the state-of-the-art SSFSC methods. In Figure 3, we show the effect of using different numbers of re-training steps (i.e. varying $m$ in Figure 2).

**Overview for two datasets with FSC methods**. In the upper part of Table 1, we present SSFSC results on miniImageNet. We can see that LST achieves the best performance for the 1-shot (70.1%) setting, compared to all other FSC methods. Besides, it tackles the 5-shot episodes with an accuracy

| Few-shot Learning Method | | Backbone | miniImageNet (test) | |
|---|---|---|---|---|
| | | | 1-shot | 5-shot |
| *Data augmentation* | Adv. ResNet, [15] | WRN-40 (pre) | 55.2 | 69.6 |
| | Delta-encoder, [29] | VGG-16 (pre) | 58.7 | 73.6 |
| *Gradient descent* | MAML, [3] | 4 CONV | $48.70 \pm 1.75$ | $63.11 \pm 0.92$ |
| | Bilevel Programming, [5] | ResNet-12$^{\diamond}$ | $50.54 \pm 0.85$ | $64.53 \pm 0.68$ |
| | MetaGAN, [42] | ResNet-12 | $52.71 \pm 0.64$ | $68.63 \pm 0.67$ |
| | adaResNet, [19] | ResNet-12$^{\ddagger}$ | $56.88 \pm 0.62$ | $71.94 \pm 0.57$ |
| | LEO, [27] | WRN-28-10 (pre) | $61.76 \pm 0.08$ | $77.59 \pm 0.12$ |
| | MTL, [32] | ResNet-12 (pre) | $61.2 \pm 1.8$ | $75.5 \pm 0.9$ |
| | MetaOpt-SVM, [10]$^{\dagger}$ | ResNet-12 | $62.64 \pm 0.61$ | $78.63 \pm 0.46$ |
| **LST (Ours)** | *recursive, hard, soft* | ResNet-12 (pre) | $\textbf{70.1} \pm 1.9$ | $\textbf{78.7} \pm 0.8$ |

| Few-shot Learning Method | | Backbone | tieredImageNet (test) | |
|---|---|---|---|---|
| | | | 1-shot | 5-shot |
| *Gradient descent* | MAML, [3] (by [13]) | ResNet-12 | $51.67 \pm 1.81$ | $70.30 \pm 0.08$ |
| | LEO, [27] | WRN-28-10 (pre) | $66.33 \pm 0.05$ | $81.44 \pm 0.09$ |
| | MTL, [32] (by us) | ResNet-12 (pre) | $65.6 \pm 1.8$ | $78.6 \pm 0.9$ |
| | MetaOpt-SVM, [10]$^{\dagger}$ | ResNet-12 | $65.99 \pm 0.72$ | $81.56 \pm 0.53$ |
| **LST (Ours)** | *recursive, hard, soft* | ResNet-12 (pre) | $\textbf{77.7} \pm 1.6$ | $\textbf{85.2} \pm 0.8$ |

$^{\diamond}$Additional 2 convolutional layers   $^{\ddagger}$One additional convolutional layer
$^{\dagger}$Using 15-shot training samples on every meta-train task.

Table 1: The 5-way, 1-shot and 5-shot classification accuracy (%) on miniImageNet and tieredImageNet datasets. "pre" means pre-trained for a single classification task using all training datapoints. Note that this is a reference table to show how much we gain by considering unlabeled data.

| | | mini | | tiered | | mini w/$\mathcal{D}$ | | tiered w/$\mathcal{D}$ | |
|---|---|---|---|---|---|---|---|---|---|
| | | 1(shot) | 5 | 1 | 5 | 1 | 5 | 1 | 5 |
| fully supervised (upper bound) | | 80.4 | 83.3 | 86.5 | 88.7 | - | - | - | - |
| no meta | *no selection* | 59.7 | 75.2 | 67.4 | 81.1 | 54.4 | 73.3 | 66.1 | 79.4 |
| | *hard* | 63.0 | 76.3 | 69.8 | 81.5 | 61.6 | 75.3 | 68.8 | 81.1 |
| | *recursive,hard* | 64.6 | 77.2 | 72.1 | 82.4 | 61.2 | 75.7 | 68.3 | 81.1 |
| meta | *hard* ($\Phi_{ss}, \theta'$) | 64.1 | 76.9 | 74.7 | 83.2 | 62.9 | 75.4 | 73.4 | 82.5 |
| | *soft* | 62.8 | 75.9 | 73.1 | 82.8 | 61.1 | 74.6 | 72.1 | 81.7 |
| | *hard,soft* | 65.0 | 77.8 | 75.4 | 83.4 | 63.7 | 76.2 | **74.1** | 82.9 |
| | *recursive,hard,soft* | **70.1** | **78.7** | **77.7** | **85.2** | 64.1 | **77.4** | 73.5 | 83.4 |
| | *mixing,hard,soft* | 66.2 | 77.9 | 75.6 | 84.6 | **64.5** | 76.5 | 73.6 | **83.8** |
| Masked Soft $k$-Means *with* MTL | | 62.1 | 73.6 | 68.6 | 81.0 | 61.0 | 72.0 | 66.9 | 80.2 |
| TPN *with* MTL | | 62.7 | 74.2 | 72.1 | 83.3 | 61.3 | 72.4 | 71.5 | 82.7 |
| Masked Soft $k$-Means [24] | | 50.4 | 64.4 | 52.4 | 69.9 | 49.0 | 63.0 | 51.4 | 69.1 |
| TPN [13] | | 52.8 | 66.4 | 55.7 | 71.0 | 50.4 | 64.9 | 53.5 | 69.9 |

Table 2: Classification accuracy (%) in ablative settings (middle blocks) and related SSFSC works (bottom block), on miniImageNet ("mini") and tieredImageNet ("tiered"). "fully supervised" means the labels of unlabeled data are used. "w/$\mathcal{D}$" means using unlabeled data from 3 distracting classes that are **excluded** in the support set [13, 24]. The results of using a small unlabeled set, 5 per class [24], are given in the supplementary materials.

of 78.7%. This result is slightly better than 78.6% reported by [10], which uses various regularization techniques like data augmentation and label smoothing. Compared to the baseline method MTL [32], LST improves the accuracies by 8.9% and 3.2% respectively for 1-shot and 5-shot, which proves the efficiency of LST using unlabeled data. In the lower part of Table 1, we present the results on tieredImageNet. Our LST performs best in both 1-shot (77.7%) and 5-shot (85.2%) and surpasses the state-of-the-art method [10] by 11.7% and 3.6% respectively for 1-shot and 5-shot. Compared to MTL [32], LST improves the results by 12.1% and 6.6% respectively for 1-shot and 5-shot.

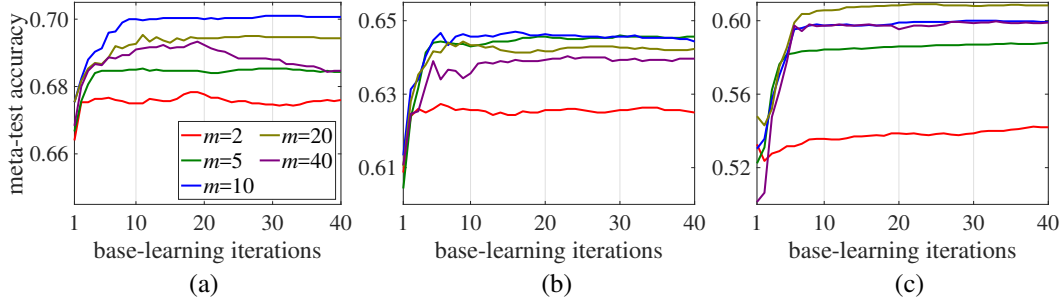

Figure 3: Classification accuracy on 1-shot miniImageNet using different numbers of re-training steps, e.g. $m = 2$ means using 2 steps f re-training and 38 steps (40 steps in total) of fine-tuning at every recursive stage. Each curve shows the results obtained at the final stage. Methods are (a) our LST; (b) *recursive, hard* (no meta) with MTL [32]; and (c) *recursive, hard* (no meta) simply initialized by pre-trained ResNet-12 model [32]. Results on tieredImageNet are given in the supplementary.

**Hard selection**. In Table 2, we can see that the hard selection strategy often brings improvements. For example, compared to *no selection*, *hard* can boost the accuracies of 1-shot and 5-shot by 3.3% and 1.1% respectively on miniImageNet, 2.4% and 0.4% respectively on tieredImageNet. This is due to the fact that selecting more reliable samples can relieve the disturbance brought by noisy labels. Moreover, simply repeating this strategy (*recursive,hard*) brings about 1% average gain.

**SWN**. The meta-learned SWN is able to reduce the effect of noisy predictions in a soft way, leading to better performance. When using SWN individually, *soft* achieves comparable results with two previous SSFSC methods [24, 13]. When using SWN in cooperation with hard selection (*hard,soft*) achieves 0.9% improvement on miniImageNet for both 1-shot and 5-shot compared to $hard(\Phi_{ss}, \theta')$, which also shows that SWN and the hard selection strategy are complementary.

**Recursive self-training**. Comparing the results of *recursive,hard* with *hard*, we can see that by doing recursive self-training when updating $\theta$, the performances are improved in both "meta" and "no meta" scenarios. E.g., it boosts the results by 5.1% when applying recursive training to *hard,soft* for miniImageNet 1-shot. However, when using *mixing,hard,soft* that learns all unlabeled data without *recursive*, the improvement reduces by 3.9%. These observations show that recursive self-training can successfully leverage unlabeled samples. However, this method sometimes brings undesirable results in the cases with distractors. E.g., compared to *hard*, the *recursive,hard* brings 0.4% and 0.5% reduction for 1-shot on miniImagenet and tieredImagenet respectively, which might be due to the fact that disturbances caused by distractors in early recursive stages propagate to later stages.

**Comparing with the state-of-the-art SSFSC methods**. We can see that Masked Soft $k$-Means [24] and TPN [13] improve their performances by a large margin (more than 10% for 1-shot and 7% for 5-shot) when they are equipped with MTL and use more unlabeled samples (100 per class). Compared with them, our method (*recursive,hard,soft*) achieves more than 7.4% and 4.5% improvements respectively for 1-shot and 5-shot cases with the same amount of unlabeled samples on miniImagenet. Similarly, our method also surpasses TPN by 5.6% and 1.9% for 1-shot and 5-shot on tieredImagenet. Even though our method is slightly more effected when adding distractors to the unlabeled dataset, we still obtain the best results compared to others.

In order to better understand our method and validate the robustness, we present more in-depth results regarding the key components, namely re-training steps, distracting classes, pseudo labeling accuracies, and using different architectures as backbone, in the following texts.

**Number of re-training steps**. In Figure 3, we present the results for different re-training steps. Figure 3(a), (b) and (c) show different settings respectively: LST; *recursive,hard* that uses the off-the-shelf MTL method; and *recursive,hard* that replaces MTL with pre-trained ResNet-12 model. All three figures show that re-training indeed achieves better results, but too many re-training steps may lead to drifting problems and cause side effects on performance. The first two settings reach best performance at 10 re-training steps while the third one needs 20 re-training steps. That means MTL-based methods (LST and the *recursive,hard*) achieve faster convergence compared to the one directly using pre-trained ResNet-12 model.

**Quantitative analyses on the number of distracting classes**. In Figure 4, we show the effects of distracting classes on our LST and related methods (improved versions *with* MTL) [24, 34]. More

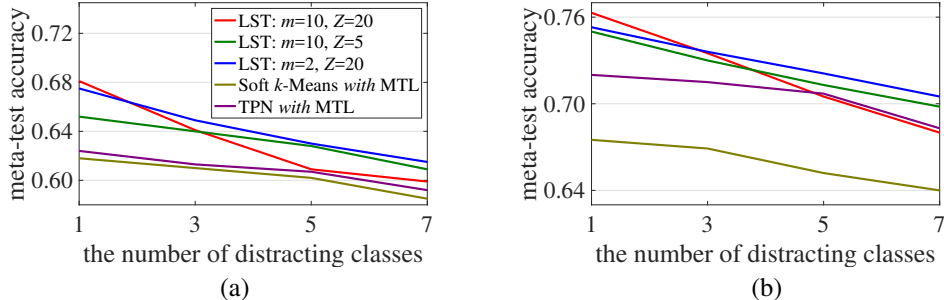

Figure 4: Classification accuracy on miniImageNet 1-shot (a) and tieredImageNet 1-shot (b), using different numbers of distracting classes.

distracting classes cause more performance deduction for all methods. Our LST achieves the top performance, especially more than $2\%$ higher than TPN [13] in the hardest case with 7 distracting classes. Among our different settings, we can see that LST with less re-training steps, i.e., a smaller $m$ value, works better for reducing the effect from a larger number of distracting classes.

**The performance of pseudo-labeling.**. Taking the miniImageNet 1-shot as an example, we record the accuracy of pseudo labeling for meta-training and meta-test (based on our best model *recursive, hard, soft*), in Table 3 and Table 4, respectively. In meta-training, we can see the accuracy grows from $47.0\%$ (iter=0) to $71.5\%$ (iter=15$k$), and it reaches saturation after $2k$ iterations. There are 6 recursive stages during meta-test. From stage-2 to stage-6, the average accuracy of 600 meta-test tasks using our best method increases from $59.8\%$ to $68.8\%$.

| Iteration | 0 | 0.2$k$ | 0.5$k$ | 1$k$ | 2$k$ | 5$k$ | 10$k$ | 15$k$ |
|---|---|---|---|---|---|---|---|---|
| Accuracy | 47.0 | 64.1 | 65.9 | 70.0 | 71.2 | 70.9 | 71.3 | 71.5 |

Table 3: Pseudo-labeling accuracies (%) during the meta-training process, on miniImageNet 1-shot.

| Stage | 1 | 2 | 3 | 4 | 5 | 6 |
|---|---|---|---|---|---|---|
| Accuracy | 59.8 | 63.6 | 65.1 | 66.9 | 67.9 | 68.8 |

Table 4: Pseudo-labeling accuracies (%) at six recursive stages of meta-test, on miniImageNet 1-shot. Stage-1 is initialization.

**Generalization ability**. Our LST approach is in principle able to generalize to other optimization-based FSC methods. To validate this, we replace MTL with a classical method called MAML [3]. We implement the experiments of MAML-based LST (using *recursive,hard,soft*) and compare with the same 4CONV-arch model TPN [13]. On miniImagenet 1-shot, our method gets the accuracy of $54.8\%$ ($52.0\%$ for w/$\mathcal{D}$), outperforming TPN by $2.0\%$ ($1.6\%$ for w/$\mathcal{D}$). On the more challenging dataset tieredImageNet (1-shot) we achieve even higher superiority, i.e., $2.9\%$ ($2.0\%$ for w/$\mathcal{D}$).

## 6 Conclusions

We propose a novel **LST** approach for semi-supervised few-shot classification. A novel recursive-learning-based self-training strategy is proposed for robust convergence of the inner loop, while a cherry-picking network is meta-learned to select and label the unsupervised data optimized in the outer loop. Our method is general in the sense that any optimization-based few-shot method with different base-learner architectures can be employed. On two popular few-shot benchmarks, we found consistent improvements over both state-of-the-art FSC and SSFSC methods.

## Acknowledgments

This research is part of NExT research which is supported by the National Research Foundation, Prime Minister's Office, Singapore under its IRC@SG Funding Initiative. It is also partially supported by German Research Foundation (DFG CRC 1223), and National Natural Science Foundation of China (61772359, 61671289, 61771301, 61521062).

## Footnotes

*This work was done during their internships mainly supervised by Qianru.

†Corresponding authors: qianrusun@smu.edu.sg; sbzh@sjtu.edu.cn.

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
