[Supplementary Material · NeurIPS_supp.pdf]

# Learning to Self-Train for Semi-Supervised
# Few-Shot Classification

**Xinzhe Li**[1*]   **Qianru Sun**[2†]   **Yaoyao Liu**[3*]   **Shibao Zheng**[1†]

**Qin Zhou**[4]   **Tat-Seng Chua**[5]   **Bernt Schiele**[6]

[1]Shanghai Jiao Tong University [2]Singapore Management University [3]Tianjin University [4]Alibaba Group
[5]National University of Singapore [6]Max Planck Institute for Informatics, Saarland Informatics Campus

## Supplementary materials

These supplementary materials include the additional results of using different numbers of stages in the recursive training in our LST method (*recursive, hard, soft*) (§A), the supplementary results (on the tieredImageNet dataset) of Figure 3 in the main paper (§B), and the comparable results using a very limited number of unlabeled data, i.e. 5 unlabeled samples per class (§C). There are also experimental results about when our LST method is equiped with different backbones (§D).

## A   Using different numbers of recursive stages

During meta-validation, we test our method using different numbers of recursive stages, and show the results in Figure 1. We observe that the performance of our method is saturated when running after e.g. 6 stages. In experiments, we split 100 samples (per class) as the unlabeled dataset. At each recursive stage, we sample a subset, i.e. 30 for 1-shot and 50 for 5-shot. After a few stages, the model has sampled and learned all unlabeled samples, therefore, its performance gets saturated. We choose the peak values, so we use 6 stages for 1-shot and 3 for 5-shot during meta-test, on both miniImageNet and tieredImageNet.

Figure 1: Meta-validation results (classification accuracy) using different numbers of recursive stages, in the 1-shot (a) and 5-shot (b) settings on the miniImageNet dataset.

Figure 2: Classification accuracy in the 1-shot tieredImageNet, using different numbers of re-training steps, e.g. $m = 2$ means using 2 steps of re-training and 38 steps (40 steps in total) of fine-tuning at every recursive stage. Each curve shows the results obtained at the final stage. Methods are (a) our LST; (b) *recursive, hard* (no meta) with MTL [4]; and (c) *recursive, hard* (no meta) simply initialized by pre-trained ResNet-12 model [4].

## B  Using different numbers of re-training steps

In Figure 2, we report the results on tieredImageNet 1-shot, using different numbers of re-training steps, as the supplementary of Figure 3 in the main paper. The same as in Figure 3, each curve shows the results obtained at the final recursive stage. Corresponding methods are (a) our LST, (b) our ablative method $recursive, hard$ (no meta) with off-the-shelf MTL model [4], and (c) the $recursive, hard$ (no meta) that directly uses pre-trained ResNet-12 model [4]. We can observe that very few re-training steps, i.e. 2 steps, are enough for our LST model to converge to the best performance, similar to the conclusion drawn from the results on miniImageNet.

## C  Using a small number of unlabeled samples

We also consider using limited number of unlabeled samples (5 per class) in the experiments. In this setting, we evaluate our LST method (the version without *recursive* due to few unlabeled data) as well as related methods, Masked Soft $k$-Means and TPN. Note that same with Table 2 in the main paper, these related methods are equipped with MTL, i.e. using pre-trained ResNet-12 as backbone, and using more efficient meta operations (scaling and shifting) in the feature extraction part. As shown in Table 1, our method achieves the best performance compared to other two methods, on both benchmarks.

| | mini | | tiered | | mini w/$\mathcal{D}$ | | tiered w/$\mathcal{D}$ | |
|---|---|---|---|---|---|---|---|---|
| | 1(shot) | 5 | 1 | 5 | 1 | 5 | 1 | 5 |
| *hard, soft* (Ours w/o *recursive*) | 61.9 | 75.3 | 72.1 | 82.4 | 60.3 | 75.0 | 70.7 | 82.0 |
| Masked Soft k-Means [2] *with* MTL | 58.2 | 71.9 | 65.3 | 79.8 | 56.8 | 71.1 | 63.6 | 79.2 |
| TPN [5] *with* MTL | 59.3 | 71.9 | 67.4 | 80.7 | 58.7 | 70.6 | 67.2 | 80.5 |

Table 1: Classification accuracy (%) using a limited number of unlabeled samples (5 per class) on two benchmarks – miniImageNet ("mini") and tieredImageNet ("tiered"). "w/$\mathcal{D}$" means using unlabeled data from distracting classes that are **excluded** in the support set [2, 5].

## D  Generalization ability

We incorporate the 4CONV arch. of MAML [1] and the recent FSC method LEO [3] into our LST, respectively. The results are shown in Table 2. For example, on tieredImageNet 1-shot, LST-MAML-4CONV outperforms TPN-4CONV [5] by $2.9\%$ and $2.0\%$ ($w/\mathcal{D}$). LST-LEO-ResNet12 outperforms TPN-ResNet12 by $3.8\%$ and $2.8\%$ ($w/\mathcal{D}$).

|  | MAML [1] | | LEO [3] | |
|---|---|---|---|---|
|  | mini(1-shot) / D | tiered / D | mini / D | tiered / D |
| *recursive,hard,soft* | 54.8 / 52.0 | 58.6 / 55.5 | 66.0 / 63.5 | 75.9 / 74.3 |
| TPN [5] | 52.8 / 50.4 | 55.7 / 53.5 | 62.7 / 61.3 | 72.1 / 71.5 |

Table 2: 5-way, 1-shot classification accuracy (%) by replacing our base network MTL(ResNet-12) [4] with MAML(4CONV) [1] and LEO(ResNet-12) [3].

## Footnotes

*This work was done during their internships mainly supervised by Qianru.

†Corresponding authors: qianrusun@smu.edu.sg; sbzh@sjtu.edu.cn.