[Reviews · NeurIPS 2019]

Reviewer 1



The paper proposes a natural combination of two methods in FSL and SSL, (namely the (good) MTL and the (basic) self training, respectively), to address the problem of learning a classifier from few labeled and many unlabeled examples. However, the trivial composition of these methods brings almost no gain from using unlabeled samples, so an effort is made to make the self-training more robust to noise by involving an additional few-shot method (namely, the Relation Network, which is basically the Siamese network) and fine-tuning on just the labeled examples (which is allowed by the initial MAML training). The experimental results corroborate the efficiency of this method, so overall a good practical knowledge is shaped and delivered by the paper. In the light of the presented results, I wonder if the proposed soft weighting of the pseudo-labels can be used in the vanilla SSL task, composed with any method based on label propagation. Perhaps authors have some results or thoughts in this direction. One issue that bothers me in Table 2 of performance results is the low accuracy reported for the baseline methods [22] and [37]. These results are lower than the concurrent performance of methods using just the few labeled examples, without the unlabeled ones. The presented ablation study is satisfying, since the performance of the different versions of the algorithm blocks helps to understand their vitality. The performance reduction due to distracting classes, demonstrated in Table 2, is a good additional

Reviewer 2



In this paper, the authors propose a novel semi-supervised meta-learning method called learning to self-train (LST) that leverages unlabeled data and meta-learnes how to cherry-pick unlabeled data to further improve performance. I think the most prominent part of the paper is the self-traing method, which is to combine the SWN (similar to the relation network) and the meta-learning training strategy to learn the relation between the unlabeled sample with the prototypes of each category, thus reducing the noise of uncertain pseudo-labels. The following aspects of the paper are insufficient or unclear: 1. The MTL (tieredImageNet) results presented in the upper part of Table 1 are indicated that “by us”. What does that mean ? 2. It might be unfair to compare with few-shot learning methods in Tabel 1. These methods neither use unlabeled set R, nor iterates 40 times to fine-tune during test. I think the statement is not suitable that “our LST performs best in both 1-shot (77.7%) and 5-shot (85.2%) and surpasses the state-of-the-art method [10] by 11.7% and 3.6% respectively for 1-shot and 5-shot.” in LINES 240 – 241. 3. Could most of few-shot learning methods be equipped with MTL to improve performances on semi-supervised few-shot classification task? 4. The results “mini w/D” and “tiered w/D” in Table2 are not very convincing. Is it the experiments that shows your method is unstable on semi-supervised few-shot learning, so that the best module selections vary greatly on different datasets? There are little explaination about the usage scenarios of “+recursive” and “+mixing”, and no theoretical proof for the explanation of LINES 259 – 261. 5. I think the method might not work well in reducing the effect of noisy labels (LINE 137) if using a lot of unlabeled data from distracting classes that are excluded in the support set and small samples from classes included in the support set. What is role of unlabeled data from distracting classes in the loss function Formula 5 (after LINE 157) ? Is it only “hard selection” strategy to reduce the impact of “false sample” from distracting classes? In summary, I think that the idea of the paper is a little bit of interesting, but contributes little. Because the scenario of few-shot learning problems is quite different from the experimental setup in the paper. The MTL method in the paper requires a large number of unlabeled samples, and iterations a lot to adjust the parameters during test. On the semi-supervised few-shot learning problems, I think that the method needs to focus on this situation when there is a big difference between the unlabeled sample categories and the supported sample categories. But I think the paper is lacking in innovation to solve this problem.

Reviewer 3



1. In the proposed training scheme, training on the data points with pseudo labels is followed by finetuning the model only on the labeled data. What will the model performance look like if finetuning on the labeled data is not used? 2. For the experimental results in Table 1, it seems only ResNet-12(pre) is used for the proposed method. What about other backbones? E.g., 4 CONV, which is used in the previous literature. Also, the comparison does not seem to include more recent approaches, e.g., RelationNet, dynamic few-shot visual learning without forgetting, etc. It would be nice to see a more extensive comparison with the previous approaches. 3. The cherry picking step is composed of a hard-selection and a soft-weighting step. What are the detailed statistics about how many unlabeled data points are filtered in each step? Also, what is the ratio of unlabeled data and the labeled data used during training? --- After rebuttal The feedback is satisfactory. I increase my score to 6.

[Author Response · NeurIPS 2019]

**R2's Q1: In vanilla SSL task.** Sorry that we don't have this experiment. Our thought is that if taking pseudo-labels
as noisy labels, perhaps one may refer to the supervised re-weighting methods [A][B]. **Q2: One issue of results[37],**
**52.8<53.8.** This is because the training splits of dataset are different between SSFSC and FSC (see details in the Sec.
4.4 of [37]). On the FSC dataset, a big proportion of labeled data are used as unlabeled for sampling SSFSC tasks.
Therefore, the total SSFSC training tasks contain less supervision (than FSC). **Q3: State of the art SSL methods.**
Actually, we began our project by trying Virtual Adversarial Training (VAT) [15] which has been shown top-performing
in most settings of vanilla SSL [18]. We found that VAT brings limited improvement, e.g. less than $1\%$ on miniImageNet
1-shot, and it works slightly better for 5-shot. We think this is because of the high-variance of FSC classifiers trained
with very limited supervision. In contrast, our method can greatly increase this supervision by carefully choosing
and weighting high-confidence pseudo labels, and thus can make a visible improvement, e.g., $9\%$ over the supervised
baseline on miniImageNet 1-shot. We agree that distracting class is a challenge (kindly refer to **R3**'s **Q5**). One of our
future works is to find out an effective way of deploying regularization-based SSL methods to tackle this.

**R3's Q1: "by us".** It means we implement the open-sourced MTL code on the
tieredImageNet. **Q2: Comparing with FSC.** We agree this is unfair in terms of (1)
the additional unlabeled data in each single SSFSC task or (2) the muted labels on
the whole dataset (kindly refer to **R2**'s **Q2**). For paper revision, we will preserve only
the comparison to baseline supervised methods and remove others. We will add more
results related to SSFSC, e.g., Figure B1. **Q4: Vary module selections: "+recursive"**
**or "+mixing".** Sorry for the confusion. "+recursive" and "+mixing" are actually
the same method (LST) with different hyperparameters, not different modules. E.g.
in 5-way, 1-shot setting, "+recursive" has 6 recursive stages, and every stage it uses
$5 \times 30$ unlabeled samples. While, "+mixing" has only one stage, using $6 \times 5 \times 30$
samples for once. **Q5: Quantitative analysis for the number of distracting classes.**
Our experiment results in Figure B1 show that both our LST and related methods,
Soft $k$-Means [22] and TPN [37], are obviously affected by distracting classes. Other
observations are that (1) LST achieves top performances, especially more than $2\%$
higher than TPN [37] at $classNum = 7$; (2) LST with less re-training steps, i.e., a
smaller $m$ value, works better for reducing the effect from a larger number of distracting
classes. **Q6: Require a large number of unlabeled samples.** In the supplementary
Table S1, we provided the results of using $5$ unlabeled samples (LINE 22-30) for both
our LST (w/o *recursive*) and related methods [22][37], validating our superiority in the
low-data settings. Note that in the Table 2 of the main paper, we reported the results of
LST (*recursive,hard,soft*) and related works using the same number of unlabeled data.

Figure B1: The 5-way, 1-shot results using different numbers of distracting classes.

**Q7: Distracting classes in Formula 5. SWN for distracting classes.** (1) Samples from distracting classes are mixed
with other unlabeled data without distinction, thus have no special role in Formula 5. (2) SWN does reduce the effect
of distracting classes. When comparing "*recursive,hard*" to "*recursive,hard,soft*" in Table 2 (w/$\mathcal{D}$), we can see the
improvements ($2.3\% \sim 5.2\%$) (*soft* = using SWN). **Q8: Accuracy of pseudo label.** Taking the miniImageNet 1-shot as
an example, during meta-training episodes, we can see the accuracy growing from $47.0\%$ (iter=0) to $71.5\%$ (iter=15$k$).
There are 6 recursive stages during meta-test. From stage-1 to stage-6, the average accuracy (of 600 meta-test episodes)
increases from $63.6\%$ ($62.2\%$ w/o *soft* weighting) to $68.8\%$ ($66.1\%$ w/o *soft* weighting). Detailed numbers will be
reported in our paper. **Q9: Insufficient aspect.** LST has some discrete hyperparameters (e.g., the numbers of hard
selected samples and recursive stages) that are manually set. Our future work is to make them optimizable.

**R4's Q2 (R3's Q3): MTL helps SSFSC.** MTL transfers the superior pre-trained DNN weights for efficient feature
extraction in unseen classes. It is independent from the learning method, either supervised or semi-supervised, for
base classifiers. Our implementations of MTL in three methods, [22][37] and ours, validate its efficiency for SSFSC.
**Q1: Without finetuning.** Please kindly refer to Figure 3(a). "$m = 40$" means the number of re-training steps is equal
to the number of total steps (40), i.e., without finetuning step. Its corresponding curve clearly drops after the 18-th
iteration. **Q3: Using other backbones/FSC approaches.** We incorporate the 4CONV arch. of MAML [3] and the
recent FSC method LEO [25] into our LST, respectively. For example, on tieredImageNet 1-shot, LST-MAML-4CONV
outperforms TPN-4CONV[37] by $2.9\%$ and $2.0\%$(w/$\mathcal{D}$). LST-LEO-ResNet12 outperforms TPN-ResNet12 by $3.8\%$
and $2.8\%$(w/$\mathcal{D}$). Other results will be reported in the final paper. **Q3: Sample statistics.** For example, in 5-way,
1-shot case, we use 1 labeled and 20 unlabeled samples *per class* to meta-train SWN. In each meta-test task, we have 6
recursive (base-)training stages. At each stage, we select 100 samples (globally ranked by pseudo-labeling confidences)
out of $5 \times 30$ unlabeled inputs, and then weight them by the SWN. If using distracting classes, we simply add 30
samples *per distracting class* to the input, without distinction. Please kindly refer to LINE 193-199 for more details.

[A] Ren et al. Learning to Reweight Examples for Robust Deep Learning. *ICLR'18.* [B] Jiang et al. MentorNet: Learning
Data-Driven Curriculum for Very Deep Neural Networks on Corrupted Labels. *ICML'18.*


[Meta-Review · NeurIPS 2019]

The reviewers overall feel that this is a good adaptation of self-training to the meta learning setup for semi-supervised few-shot learning. Many of the reviewer concerns were addressed by the additional experiments and clarifications provided in the rebuttal. Please include these in the final draft.